# Investigation of *Streptomyces*
*scabies* Causing Potato Scab by Various Detection Techniques, Its Pathogenicity and Determination of Host-Disease Resistance in Potato Germplasm

**DOI:** 10.3390/pathogens9090760

**Published:** 2020-09-17

**Authors:** Sohaib Ismail, Bo Jiang, Zohreh Nasimi, M. Inam-ul-Haq, Naoki Yamamoto, Andrews Danso Ofori, Nawab Khan, Muhammad Arshad, Kumail Abbas, Aiping Zheng

**Affiliations:** 1Department of Plant Pathology, Sichuan Agricultural University, Chengdu 611130, China; sohaibismail@stu.sicau.edu.cn (S.I.); z.nasimi@sicau.edu.cn (Z.N.); 80053@sicau.edu.cn (N.Y.); andrewsdanso@icloud.com (A.D.O.); 2College of Lifescience and Technology, Yangtze Normal University, Chongqing 408100, China; 20050010@yznu.edu.cn; 3Department of Plant Pathology, PMAS-Arid Agriculture University, Rawalpindi 44000, Pakistan; dr.inam@uaar.edu.pk; 4Department of Agricultural Economics, Sichuan Agricultural University, Chengdu 611130, China; nawabkhan@stu.sicau.edu.cn; 5Department of Microbiology, Sichuan Agricultural University, Chengdu 611130, China; Arshadarid2@gmail.com; 6Institute of Horticulture, Sichuan Agricultural University, Chengdu 611130, China; kumailabbas815@gmail.com

**Keywords:** *Streptomyces scabies*, common scab disease, Gram-positive bacteria, soil borne pathogen

## Abstract

*Streptomyces scabies* is a Gram-positive bacterial pathogen that causes common scab disease to several crops, particularly in the potato. It is a soil borne pathogen, a very devastating scab pathogen and difficult to manage in the field. *Streptomyces* has several species that cause common scab such as *S. scabiei, S. acidiscabies, S. europaeiscabiei, S. luridiscabiei, S. niveiscabiei, S. puniciscabiei, S. reticuliscabiei, S. stelliscabiei, S. turgidiscabies*, *S. ipomoeae*. Common scab disease harmfully affects potato economic and market value due to the presence of black spots on the tuber. Owing to its genetic diversity and pathogenicity, the determination of pathogen presence in potato fields is still challenging. In this study, *S. scabies* genetic diversity was measured by surveying five potato-growing areas of Pakistan during the growing season 2019. A total of 50 *Streptomyces* isolates, including *S. scabies*, *S. acidiscabies*, *S. griseoflavus* were isolated and identified based on morphologic, biochemical and molecular analysis. Virulent confirmation assays confirmed ten virulent strains of *Streptomyces* spp. On the potato cultivars Cardinal and Santee. Among the *Streptomyces* species, *S. scabies* showed the highest scab index, followed by *S. acidiscabies* and *S. griseoflavus* by exhibiting the scab-like lesions on potato tubers. Ten potato cultivars were screened against these virulent isolates of *Streptomyces*. The Faisalabad white variety showed the highest scab index followed By Cardinal, Tourag, Kuroda, Santee, Lady Rosetta, Asterix, Diamant, Faisalabad red and Sadaf. Moreover, genetic diversity and pathogenicity of *Streptomyces* spp. on potato tubers were also likely diverse in different geographical regions and also potato cultivars. This study represents a contribution to understanding the local interaction between potatoes and *Streptomyces* spp. in Pakistan. It will aid in supporting a solution for the management of this pathogen around the world.

## 1. Introduction

The potato (*Solanum tuberosum* L.) is designated as the sixth most imperative commodity in agriculture worldwide after sugarcane, wheat, rice, maize and cereals. The origin of the potato was Peru (South America) and transfer to the rest of the world by shipment, transportation and war expeditions [1]. In Pakistan, the potato is cultivated over an area of 0.15 million hectares, with an annual production of 2.9 million tons [2]. The potato is grown in three seasons; spring, summer and autumn are ranging from plain to hilly agro-economical areas in Pakistan [3]. The potato is the main staple food of many countries, which comprises many nutrients, proteins, carbohydrates, vitamins, minerals, fiber contents and energy [4]. In the world, China is the largest potato producing country with 99.5 million tons production per year with a production share of 25.02%, followed by other countries such as Russia, India and the United States [5].

Despite its food security significance and great market value, the potato crop is susceptible to many ailments caused by bacteria, viruses, nematodes and fungi [6,7,8]. Among bacterial diseases, the common scab is the most devastating disease of the potato, which causes economic losses to the potato-growing countries of the world [9,10]. This disease is caused by *Streptomyces scabies*, which belongs to the phylum actinobacteria, which is one of the biggest taxonomic units among the 18 major lineages of bacteria and its divergence from other bacterial species is so ancient that it is currently not possible to identify their most closely related group [11]. Arguably, actinobacteria’s best-studied genus is *Streptomyces*, which have complex developmental life cycles [12,13,14]. Potato scab disease, which is caused by *Streptomyces scabies* has been reported in many countries such as China [15], South Africa [16], Pakistan [17], Iran [18], Russia [19], India, United States [20] and several other countries of the world. This disease includes many symptoms such as raised, deep pitted, sunken lesions and scab like surface on the tuber [21]. *Streptomyces* spp. Many economical and medicinal values, such as two-thirds of antibiotics, are developed from Actinomycetes worldwide, and about 80% of antibiotics are developed from *Streptomyces* spp. [22,23]

The key stages of the *S. scabies* are the germination of spores like fungi and the outgrowth of mycelia for substrate feeding. Despite many nutritional and other stresses, this actinobacteria can germinate aerial reproductive hyphae, which reproduce many cell division *spores. Streptomyces* spp. produce many antibiotics against other pathogens, but these chemicals are produced at the stationary stage [24,25]. *Streptomyces* sp. interacts directly as well as indirectly with the plant, has positive (biocontrol of many plant pathogens, biofertilization) and negative (plant diseases like the common scab of potato) impacts on the health of the plant [26,27,28]. However, many species of *Streptomyces* such as *S. scabies*, *S. ipomoeae*, *S. turgidiscabies* and *S. acidiscabies* cause many symptoms on several hosts that include deep pitted and raised scab-like lesions on potato, beet, radish and peanut crops. These crops are economically important, but they reduce these crops’ market and consumption values [29,30].

Potato scab disease is transmissible from seed and soil sources [31,32]. The disease develops when the tuber starts emerging in the first growth stages of tubers when enlarges or direct penetration to the epidermis and enlarging the potato tuber [33]. Programmed cell death occurs near the diseased areas of tubers. Then these spots/lesions consequently transfer into deep pitted shallow lesions due to the bacterization of nearby tuber areas, which are the initial symptom development of the disease. These lesions, which develop on tubers, are circular when they are multiple; these merge to develop asymmetrical scabby lesions [34,35]. Some other factors can affect the production of the potato, such as the unavailability of seed, poor quality seed and management problems [36,37]. All the above factors affect potato yield, but among biotic factors, the factors that cause the most severe damage are diseases. The potato crop is susceptible to black scurf, powdery scab (*Spongospora subterranea*), wilt (*Verticillium albo-atrum*), but highly susceptible to common scab (*Streptomyces scabies* [38,39] which took place in the family Streptomycetaceae. *Streptomyces* spp. are the source for the production of numerous antibiotics; among the most important of these are streptomycin from (*S. griseus*), tetracycline (*S. rimosus*), neomycin (*S. fradiae*), daptomycin (*S. oseosporus*), chloramphenicol (*S. venezuelae*), lincomycin (*S. lincolnesis*), fosfomycin (*S. radial*), oleandomycin and boromycin (*S. antibioticus*), mycangimycin (*Streptomyces* spp. SPB74), tunicamycin (*S. orulosus*) and puromycin (*S. alboniger*) [40,41].

One of the best control strategies to manage this disease is resistant cultivars because *Streptomyces* bacteria have the ability of genetic variation [36,42]. Some natural and physical barriers like stomatal compactness, condensed skin, skin color, and structures are key factors in disease suppression [29]. The factors that are involved in the outbreak of the disease, are susceptible potato varieties, suitable soil factors, the existence of pathogen pathogenicity [29]. A broad range of control measures is present, but the most efficient and economical strategy to control the disease is resistant to cultivars/varieties. One of the best methods to control this pathogen is through resistance [43]. Still, the Host immunity to disease-causing microorganisms is established on the plant genetics and the continuous association between the host plant and pathogen [44,45].

Although the significance of *Streptomyces* species to potato crop was documented in previous studies, studies regarding the pathogenicity, characterization and genetic diversity are still challenging and poorly investigated in potato crops [46,47]. For this purpose, this study was conducted to check the pathogenicity and also disease resistance in potato germplasm. This study’s main objectives were to check the isolation and purification of *Streptomyces scabies* and screen the potato germplasm against *Streptomyces scabies* in greenhouse conditions and check the disease resistance in potato-growing areas of Punjab, Pakistan. Additionally, current work will help provide the information and knowledge for the resistance against common scab disease and disease-free cultivars and disease management in Punjab, Pakistan—as well as around the world.

## 2. Results

### 2.1. Diversity and Isolation Frequency of Streptomyces Species from Different Regions

Different major potato-growing areas viz: Taxila, Hazro and districts of central Punjab (Pakistan) Sialkot, Okara, Sahiwal, Kasur and Faisalabad were surveyed for the collection of infected samples. From each city, 10 fields were selected. Various disease parameters such as disease percent incidence and percent scab index were also measured during the survey. Among the disease surveyed areas, Okara showed a maximum disease incidence of 44.33%, followed by Kasur, Sialkot, Faisalabad, Taxila and Hazro, which showed disease percent incidence of 40%, 28%, 27.33%, 8% and 7%, respectively. The maximum percent scab index was observed in Kasur 22.5%, followed by Okara, Faisalabad, Sialkot, Hazro and Taxila, which exhibited a scab index of 22.16%, 16.23%, 16.14%, 4.91%, 4.42% correspondingly (Table 1, Figure 1). During the survey, the temperature was observed between 30 and 35 °C and humidity 75–80%. Common scab is the major disease of the field also causes many losses in the storage conditions after harvest. Therefore, the use of resistant varieties in the major management strategy controls the disease in the potato-growing countries of the world such as China, United States, Russia, India, South Africa [48].

As shown in Figure 2, the isolation frequency of each identified *Streptomyces* species associated with the potato tubers was different. Among them, *S. scabies* and *S. acidiscabies* had the highest isolation frequencies, i.e., 47% and 25%, respectively, followed by *S. turgidiscabiei*, *S. griseoflavus*, and *S. europaeiscabiei* had the lowest among the identified *Streptomyces* species, i.e., 12%, 10% and 4%, respectively. Thus, *S. scabies* was the dominant *Streptomyces* species associated with potato tubers, causing common scab in Pakistan’s various potato-growing areas. Samples collected from various showed, Taxila, Hazro, Sialkot, Okara, Faisalabad reported maximum pathogenicity of *S. scabies* followed by *S. tugidiscabiei*, *S. acidiscabiei, S. griseoflavus* and S. europaeiscabiei, as shown in Figure 3. In the many other countries of the world, such as China [15], South Africa [16], Pakistan [17], Iran [18], Russia [19], India, the United States, *S. scabies* is also the predominant species among all *Streptomyces* species [20].

### 2.2. Phenotypic Characterization

In the current study, 50 *Streptomyces* isolates were isolated from 175 disease tubers from 10 potato varieties (Table 2). *Streptomyces* spp. were initially identified based on morphologic characteristics, which are described in Table 3. Isolation was carried out by the serial dilution method, and plates were kept in an incubator at 28 ± 2 °C. Single colonies with different cultural characteristics were streaked on the YMA medium and incubated at 28 ± 2 °C for about 3–7 days from each culture plate. The purified cultures were obtained by re-isolating; general colony characters were observed to confirm *Streptomyces* strains based on morphology, biochemical and cultural characteristics [48].

On yeast malt agar, 23 isolates of *S. scabies* were retrieved, and they exhibit white to creamy white and abundant growth habit as *S. scabies* belongs to class Thallobacteria. It produces fungal-like hyphal growth on culture media. On isolation, it develops branched and slender spores, which are coenocytic (aseptate) and have no cross walls. The diameter of the hyphae is half to two micrometers. When spores are mature, the hyphae turn into three aerial mycelial hyphae. A total of 20 isolates of *S. scabies* were isolated on nutrient agar media. It develops small bacterial colonies with a small size in few days of incubation and later into turns into the smooth mycelial surface of actinomycetes, which appears as powdery, fluffy and granular spores. A total of 20 isolates of *Streptomyces* spp. were isolated on KB media and showed fluffy white growth.

A total 23 isolates CS-TAX01, CS-TAX002, CS-TAX004, CS-TAX08, CS-TAX13, CS-HAZ14, CS-HAZ62, CS-SIAL060, CS-SIAL70, CS-SIAL72, CS-SIAL74, CS-SIAL76, CS-OK0101, CS-OK120, CS-OK124, CS-OK126, CS-OK128, CS-OK130, CS-FSD139, CS-FSD149, CS-FSD0149, CS-FSD143, CS-FSD146 were confirmed as *S. scabies.* The growth and cultural characteristics of *Streptomyces* were checked on different solid media. On nutrient agar media, the growth was observed moderate, mycelial growth was abundant and colony color was creamy white. On potato dextrose agar, the same growth characteristics were observed. On yeast malt agar, abundant mycelial growth was observed. On KB media, poor mycelial growth, whitish to creamy colony color was observed. On oatmeal agar, the same growth as nutrient agar was observed, but colony color was whitish. Potato yeast extract showed brownish colony color and czapek media showed whitish colony color (Table 4). The bacterial (actinomycetes) colonies are comprising creamy to whitish rings with yellow margins. These cultures exhibited a fluffy to a slightly fluffy texture. The hyphal growth was slightly raised. The minimum colony diameter was estimated at 55 mm, while the maximum was 67.5 mm in seven days. The spores were two-celled and hyaline. Their shape was pointed from one end and blunted from the other end. The minimum length of the spore was estimated at 12 µm, and the maximum was 15.5 µm. The minimum width was 5.5 µm, while the maximum was 6.5 µm. These measurements were in close association with taxonomic keys with very few differences [49,50,51].

For the biochemical characterization of *Streptomyces* species, biochemical tests such as gram staining test, catalase oxidase test, Loop test, Kovac’s oxidase test were performed. The objectives of these tests were to differentiate between Gram-positive and Gram-negative bacteria. Out of 20 isolates, 12 isolates changed their colorization to purple, which were confirmed as Gram-positive bacteria, and eight isolates did not change their color and were designated Gram-negative bacteria (Figure 4 and Figure 5).

A catalase oxidase test was also performed to check the differentiation between Gram-positive and Gram-negative bacteria. Fourteen isolates make gaseous bubble formation during this process, which were designated as *Streptomyces* spp. and six isolates did not form any bubble formation where were confirmed as Gram-negative bacteria. The development of gaseous bubbles confirms the aerobic nature of bacteria, also known as facultative anaerobic bacteria [52]. A loop test was also performed to check the presence of *Streptomyces* spp. In addition, 20 isolates were taken and treated with thee percent potassium hydroxide solution. A total of 13 isolates formed a loop during this test showed that 13 isolates did not disrupt the cell membrane and show manifestation to dislocate the cell membrane, which was confirmed as *Streptomyces* spp. The other seven isolates did not form any loop which disrupts the cell membrane of bacteria. Kovac’s oxidase test was also performed in which isolates were treated with oxidase reagent impregnated on filter paper that develops a purple color. Isolates developed various colors on filter paper. Of 20 isolates, 15 isolates developed a purple color, which confirmed as Kovac’s oxidase-positive. The other five isolates formed a light purple color, which was designated as Kovac’s oxidase negative. When isolates were treated with Kovac’s oxidase solution on filter paper, they developed a purple color [52]. Hence, our results were confirmed according to the methodology of Kovac’s oxidase test, as shown in Figure 5.

### 2.3. Pathogenicity Tests/Virulence Confirmation Assays

Pathogenicity tests/virulence confirmation assays were carried out on two potato cultivars viz: cardinal and Sante. Infection was taken after 30 days of sowing (DAS) for confirmation. Re-isolation of *Streptomyces* spp. was carried out from infected potato tubers. Data collected 30 days after sowing from potato tubers revealed that among tested 10 tested isolates, three isolates CS-TAX02, CS-HAZ14, CS-HAZ56 were found highly pathogenic/virulent, four isolates CS-SIAL060, CS-SIAL101, CS-OK010, CS-OK126 were found moderately virulent and three isolates CS-OK14, CS-FSD124, CS-FSD129 were found less virulent (Figure 6).

An in vitro experiment, pinprick method was also performed to check pathogenicity, 30 potato tubers were inoculated with 10 virulent strains of *Streptomyces* species with three replications. After 24 h of inoculation, out of 30, 14 potato tubers cause a hypersensitive response, cause extensive cell death and exhibited lesions on tubers, which create resistance to other cells and 16 tubers, after three days of inoculation, showed extensive cell death and were confirmed as *Streptomyces* spp. as shown in Figure 7.

### 2.4. Screening of Potato Germplasm against S. scabies

Ten potato cultivars (Cardinal, Diamant, Santee, Faisalabad white, Faisalabad red, Asretix, Sadaf, Kuroda, Lady Rosetta, Tourag) were evaluated for screening in greenhouse as shown in Table 5, against isolates CS-TAX02, CS-HAZ14, CS-HAZ56, CS-SIAL060, CS-SIAL101, CS-OK010, CS-OK126 CS-OK14, CS-FSD124, CS-FSD129 were used for inoculation on potato germplasm. After a few days, tubers were assessed for disease severity index using a disease rating scale and disease incidence was measured using the formula and disease rating scale. Cultivars Santee and Kuroda showed high disease susceptibility against CS-TAX02 and CS-HAZ14 isolates and were identified as *S. scabies*. Tourag and Cardinal Cultivars showed moderate susceptibility against CS-HAZ56, CS-SIAL060 and CS-SIAL101 and identified species on these varieties were *S. acidiscabies*, *S. scabies*, *S. griseoflavus*. Other cultivars like Sadaf, Lady Rosetta and Diamant showed minimum disease susceptibility against CS-OK010, CS-OK126, CS-OK14 and species, which was identified as *S scabies*. The most identified and dominant species reported in the screening of potato germplasm was *S. scabies*.

### 2.5. Measurement of the Percentage of Potato Scab Incidence

Potato scab incidence was calculated based on the number of diseased tubers than healthy tubers. Asterix Cultivar showed 2.90% disease incidence, which was found resistant variety. Four cultivars Lady Rosetta, Diamant, Sadaf, Faisalabad-white, were found less susceptible cultivars, which show disease incidence of 17.55%, 41.55%, 33.38% and 22.53%, respectively. Two cultivars were found to be susceptible cultivars, Santee, Kuroda, which show percent disease incidence of 49.63% and 56.35% and percent scab index of 20.5% and 20.65%. Two cultivars Cardinal and Tourag, were found moderate susceptible in the experiment, which shows disease incidence of 37.46% and 46.25%, respectively. One cultivar Faisalabad red was found highly susceptible, which shows disease incidence of 75.24%, as shown in Figure 8. Tubers that were not inoculated with *Streptomyces scabies* were observed resistant. In this study, for the first time, screening of potato germplasm cultivars was demonstrated against *Streptomyces scabies* in Pakistan.

### 2.6. Measurement of Potato Scab Index and Percent Disease Severity

Potato cultivars; Asterix, Lady Rosetta, diamant, sadaf, Faisalabad white, Santee, Kuroda, Cardinal, Tourag and Faisalabad red showed potato percent scab index of 1.47%, 5.90%, 11.38%, 7.09%, 1.08%, 20.5% and 20.65%, 15.91%, 18.77% and 25.18%, respectively. Mean scab indices, respectively, with their mean. The scab index provides a more patent depiction of disease severity than disease incidence.

### 2.7. Effect of Streptomyces scabies on Growth Parameters of Potato Cultivars

Data of different potato parameters were collected and evaluated for statistical analysis (Table 6). Data were obtained from three independent experiments. In each experiment, three replicates were used. Cultivar Kuroda showed root weight of 3.64 g, shoot weight 9.76 g, root length of 20.66 cm, shoot length of 30.66 cm and 6.01 number of tubers followed by cultivars Santee, Faisalabad red, Tourag, Diamant, Cardinal, Sadaf, Asterix, Faisalabad white and Lady Rosetta which showed RW of 3.24, 2.22, 4.24, 4.81, 5.85, 3.56, 4.96, 2.59 and 3.81 g, respectively, shoot weight (SW) of 9.64, 6.83, 10.44, 11.54, 14.78, 10.61, 12.37, 7.36 and 10.63 g, root length (RL) of 19.66, 19.66, 21.33, 19.01, 16.01, 16.33, 20.33, 18.33 and 21.33 cm, shoot length (SL) of 39.66, 41.01, 41.33, 34.01, 40.33, 40.01, 26.66, 30.66 and 46.01 cm, No of tubers (NT) of 5.66, 5.01, 5.66, 6.01, 4.66, 5.66, 5.33, 5.66, 5.01, respectively as shown in Figure 9 and Table 7.

### 2.8. Molecular Detection of Streptomyces scabies (Gel Electrophoresis and PCR Amplification)

Universal primers for the 16 S rRNA gene were utilized for amplification of confirmed bacterial DNA in overall particular positive isolates to approve the pathogens. The polymerase chain reaction products of 1500 bp were achieved in 15 out of 17 isolates. The isolates at well 4, 10—and those used as a negative control (shown as zero (0))—did not exhibit any amplification. Same as the consequences of PCR detection were documented in Malaysia by [52]. Furthermore, polymerase chain reaction analysis conducted by [53] also detected this infection in India. In the present investigation, 10 isolates of *Streptomyces* spp. by molecular detection, were identified from the various zones of Punjab. Such isolates were also detected via biochemical investigation and further confirmed by PCR employing a particular primer (Figure 10).

### 2.9. Sequence Analysis

Ten isolates were selected from the PCR amplification procedure, and seven isolates showed similarity with *S. scabies*, which were matched accession number from the NCBI website that includes M57297.1, AM293590.1 which showed 100% similarity, HM018077.1, AY207602.1, Y15497.1 showed 99% similarity, AF031232.1, AB301479.1 showed 98% similarity and two isolates were matched with S. *europaeiscabiei* HQ441817.1, AY207595.1 *and showed the similarity of 89 and 90%, one isolate*
*S. griseoflavus *showed the similarity of 96% and matched with** JX284407.1 accession number (Table 8). Phylogenetic trees were constructed by the neighbor-joining method with the Kimura two-parameter model in the MEGA 7 version [54] with 1000 bootstrap replicates (Figure 11).

## 3. Discussion

In this study, we isolated and characterized 50 isolates of *Streptomyces* species, including *S. scabies*, *S. acidiscabies*, *S. europaeiscabiei* and *S. turgidiscabies*, from diseased samples collected from different potato-growing areas of Pakistan. It is noteworthy, these species have been recognized as the pathogens of Potato and cause common scab disease. The number of isolates were different from each growing area, respectively. Among them, *Streptomyces scabies* was the dominant species as compared to other *Streptomyces* species: out of 50 identified isolates, 23 isolates were identified as *S. scabies*. This result was similar to a study conducted by [54,55]; they also reported that *S. scabies* is the most prevalent species among all *Streptomyces* species. Although, common scab disease is caused by many *Streptomyces* species in many crops, but most prevalently in potato and carrot, but main causal pathogen of this disease is *S. scabies*. This disease does not cause many yield losses but reduces the market and economic value of these crops because the diversity of the *Streptomyces* species is very variable. These results were also reported by many previous studies conducted by [13,54]; (S) *scabies* is dominant species in many countries of the world, such as China [15], South Africa [16], Pakistan [17], Iran [18], Russia [19], India, United States [20]. Previous studies demonstrated that *S. scabies, S. acidiscabies, S. europaeiscabiei* and *S. turgidiscabies* cause common scab disease of Potato [56,57]. However, the diversity and pathogenicity of *Streptomyces* species are distinct in the current study. In this disease, the Identification of diseased tubers is conspicuous since the pathogen does not destroy the tubers, but their surface blemishes, decreasing the yield’s market value. Management of the common scab can be done by soil fumigation, chemical control, lowering soil pH and antibiotics use. However, these management practices are not only difficult and almost impossible in large fields, and the only economical method for the management of common scab is the use of resistant varieties [55]. Several biochemical tests were also performed for confirmation of gram positive actinomycetes bacteria to check either these isolates are *Streptomyces* spp. are test-positive or test-negative. Our results were compared with work done by previous studies. Ten virulent isolates were taken to perform pathogenicity tests on two potato cultivars, Cardinal and Santee. These cultivars showed less susceptibility when these isolates were applied for virulence confirmation assays. These results were also similar to work done by [54]. In Pakistan, the reports on screening of potato germplasm against common scab are very old, and recent germplasm has not been screened for the disease. Hence, a dire need to screen the available potato germplasm against *Streptomyces scabies* to manage the disease by using resistant varieties. Potato cultivars such as Cardinal, Diamant, Santee, Faisalabad white, Faisalabad red, Asretix, Sadaf, Kuroda, Lady Rosetta, Tourag were screened during 2019 to check the resistance variability. The asterix and Faisalabad white showed 1.47 and 1.08 percent scab index, designated as resistant varieties according to disease scale. Previous studies showed that Cardinal showed the least susceptibility and Santee moderate susceptibility [54]. In our findings, *S. scabies* significantly cause scab-like lesions than other species. This is demonstrated by [45], who reported that *S. scabies* was abundantly isolated by potato tubers and indicating the specific penetration ability to potato tuber; (S) *scabies* is a soilborne pathogen that penetrates the potato tuber when it has a small size and germinates outside the tuber skin. Percent disease incidence and percent scab index were measured by formulas. According to results it was observed that maximum disease incidence % was observed in cultivar Kuroda 56.48% and maximum scab index % was observed in cultivar Santee 21.16% during the screening of potato germplasm. These results were compared by work of [54] which showed that Astrax showed maximum disease incidence % of 59.80% and maximum scab index% was observed in also Astrax of 41.80%, respectively. Previous studies supported that *S. scabies* often has a close phylogenetic relationship between *S. acidiscabies, S. europaeiscabiei* and *S. turgidiscabies* [58]. Phylogenetic tree analyses and DNA sequencing revealed the genetic diversity, variability and pathogenicity in the potato-growing areas of the Pakistan and will be helpful for the management of common scab disease around the world. The phylogenetic relationship and DNA sequences were compared with study done by [59]. These all species were found on potato tubers but differed significantly in aggressiveness. It is obvious that tuber infection had a direct connection with tuber decay. However, the current may be useful for breeding resistant varieties of Potato and distinguishing the *Streptomyces* species in potato-growing areas of Pakistan.

## 4. Materials and Methods

### 4.1. Diversity and Isolation of Streptomyces Species from Different Regions

In 2019, Symptomatic tubers were collected at harvesting time from 10 potato varieties and different potato-growing areas of Punjab, Pakistan: Taxila (Rawalpindi) (33.7463° N, 72.8397° E), Hazro (Attock) (33.9053° N, 72.4791° E), Islamabad (33.6844° N, 73.0479° E), Omari (30.8138° N, 73.4534° E), Faisalabad (31.4504° N, 73.1350° E), Sialkot (32.4945° N, 74.5229° E) as shown in Figure 12.

From each variety, 10 plants were selected for collection of diseased tubers. During the sampling, the age of tuber was about one to two months after plantation of potato crop in sampling areas. The tubers were small during the sampling, because potato scab disease developed mostly in young tubers, when potato has young epidermis and has not fully developed periderm. *S. scabies* attacks mostly in immature tubers and remain until the tuber is fully developed. Samples were collected based on symptoms such as raised and deep pitted scab like lesion. During the survey, the temperature was observed between 30 and 35 °C and humidity 75–80%. For isolation of *Streptomyces* species, samples were washed with distilled water and then disinfected with a 3% Clorox solution, followed by the tubers’ rinsing with distilled for about one minute. After this, scabby lesions were cut with a sterilized scalpel and homogenized with two milliliters Tris-HCl solution (pH 7.2) in an Eppendorf tube and then incubated for 10 min at 45 °C in the incubator. After this, from the homogenized sample, 100 µL solution was taken and spread on KB agar media (peptone 20.0 g, glycerol 8.0 mL, Agar 12.0 g, K_2_HPO_4_ 1.5 g, MgSO_4_. Then, 7H_2_O 1.50 g, distilled water 1000 mL) supplemented with 50 μg∙mL^−1^ streptomycin and incubated at 25 ± 2 °C in the dark for seven days until the *Streptomyces* spore grew on culture media. Isolates were purified by picking a hyphal tip from the actively grown colonial margin and preserved on KB media for further identification. All the isolates were purified and established by single spore isolation [60].

### 4.2. Phenotypic Characterization

Eight solid media including yeast malt extract agar media (YMA), peptone yeast extract agar iron media (PYI), King’s B, Czapek’s Dox, potato dextrose agar (PDA), tyrosine–casein–nitrate agar (TCN), Oat meal agar and nutrient-agar media were used for cultural characteristics of isolated *Streptomyces* species. The growth rate and mycelial development were examined on these solid media. Three media plates were cultured for each plate, and experiments were repeated two times to check fluctuations of growth rate. For the biochemical characterization of the recovered pathogenic isolates, gram staining test, catalase oxidase test, catalase oxidase test, loop test, Kovac’s oxidase test were carried out. All tests were performed as described by [61].

### 4.3. Pathogenicity Tests/Virulence Confirmation Assays

The pathogenicity of identified *Streptomyces* species was performed on healthy potato tubers of two varieties (Cardinal and Santee). The spore suspension was prepared by transferring the 5–6 plugs of actinomycetes bacterial pathogen on 50 mL of yeast extract media without agar and incubated at an orbital rotator at 150 r.min^−1^, 28 °C for 72 h to obtain a concentration of 1 × 10^6^ CFU mL^−1^. After this, spore suspension was mixed immediately on the sterilized-silty loam soil. The potato varieties were sown on infested soil with three replications for each treatment. For control, only distilled water was flooded in soil despite the *Streptomyces* inoculum. Each pot carried the same inoculum (10^6^ CFU mL^−1^) and kept in a greenhouse with 25–30 °C temperature and 75% humidity. Symptoms were assessed after two months of sowing (DAS). Pathogenicity of identified isolates using inoculum (spore suspension) was performed to fulfill Koch’s postulates.

Pinprick method was also used to confirm the pathogenicity of the purified culture of *Streptomyces* species according to Koch’s postulates. A sterilized toothpick inserted in a vial containing a 48 h-old culture of *Streptomyces scabies* and later inserted into healthy potatoes. A toothpick inserted into sterilized water was also inserted into healthy potatoes considered as control. Two to three toothpicks were used for a single potato. The potatoes later were put into an incubator at 30 ± 2 °C to observe peculiar symptoms of common scab. Bacteria from infected potatoes re-isolated to confirm Koch’s postulates. This test proved that *Streptomyces scabies* is pathogenic to potato crop and develops cracks symptoms, which were demonstrated by pathogenicity test. When *S. scabies* was re-isolated, it developed the same growth characters.

### 4.4. Screening of Potato Germplasm against Streptomyces scabies

Healthy certified potato tubers were obtained from different sources, as shown in Table 9 and all the available potato germplasm were screened for resistance against *Streptomyces scabies*’ most virulent isolates. Aggressive virulent strains, as confirmed by pathogenicity tests, were selected for the available germplasm. The selected strains were multiplied on KB media from each strain’s glycerol stock and incubated for 3–7 days at 28 °C. Mass inoculum of the selected strains of *Streptomyces scabies* was grown in 500 mL flasks containing 250 m1 of nutrient broth. The potting mixture was prepared by mixing the sand, soil and farmyard manure in equal proportion (1:1:1) and sterilized with 37% formalin following. One of the sprouted potato with 3–4 eyes of the same size was first inoculated with each strain by dipping into broth inoculum. The inoculated tubers were placed in a depth of 4–5 cm and covered with a sterilized potting mixture. Pots were watered as required and kept at 25 °C in greenhouse conditions.

### 4.5. Percentage of Disease Incidence

Disease incidence was calculated based on the number of diseased tubers. Tubers were examined after the inoculation of *Streptomyces scabies.* Disease incidence was measured based on the disease rating scale, as shown in Figure 13. Finally, the percentage of disease incidence was calculated following the formula of percent disease incidence: $$\mathrm{Percent}\;\mathrm{disease}\;\mathrm{incidence}=\frac{\mathrm{Number}\;\mathrm{of}\;\mathrm{infected}\;\mathrm{tubers}\;}{\mathrm{Total}\;\mathrm{No}.\;\mathrm{of}\;\mathrm{tubers}}\times 100$$

### 4.6. Percentage of Potato Scab Index and Disease Severity

Disease severity was measured based on symptoms on the surface of tubers. Disease severity was recorded according to disease severity rating scales [62], as shown in Figure 14. Where,
0 = No symptom,1 = Very small lesions,2 = Small superficial lesions,3 = Periderm broken,4 = Light pitted5 = Deep pitted

The susceptibility of potato cultivars to common scab was grouped according to Marais and Vorster (1988), which is given below: The formula for scab index is: $$\mathrm{Scab}\;\mathrm{index}=\frac{\%\;\mathrm{age}\;\mathrm{surface}\;\mathrm{area}\;\mathrm{covered}}{15}\times \;\mathrm{lesion}\;\mathrm{type}\;\times \;100$$

Ten potato varieties (Cardinal, Santee, Lady Rosetta, Kuroda, Tourag, Asterix, Diamant, Faisalabad white, Faisalabad red, Sadaf) were screened in three replicates against two virulent *Streptomyces scabies* isolates. Plants that have not been inoculated with the pathogen, only inoculated with distilled, have also been kept in control conditions as a negative control.

### 4.7. Effect of Streptomyces scabies on Growth Parameters of Potato Cultivars

The plants were monitored regularly and data of disease incidence (DI), shoot length (SL), Shoot fresh weight (SFW) and shoot dry weight (SDW) were also collected.

### 4.8. Molecular Detection of S. scabies (Gel Electrophoresis and PCR Amplification)

The total genomic DNA of *Streptomyces* strains was recovered by GeneJET Genomic DNA purification Kit (Thermo Scientific, Waltham USA) as per the manufacturer’s instructions. For DNA isolation, 2 × 10^9^ bacterial cells were taken in 1.5 mL microcentrifuge tubes and centrifuged at 5000× *g* rpm for 10 min. and the supernatant was discarded. The obtained pellet was resuspended in 180 µL of digestion solution. Then, 20 µL proteinase K was added, and a uniform solution was obtained by vortexing. Samples were incubated at 56 °C in a shaking water bath till the complete bacterial cell lysis. Then, 20 µL RNase solution (A) was added to it and mixed by vortexing and incubated at room temperature for 10 min. Then, 200 µL lysis solution was added to the samples and vortexed for 15 s until a homogenous mixture is obtained, and then 400 µL of 50% ethanol was added to the mixture and vortexed. The prepared lysate was shifted to a purification column and inserted in collection tubes and centrifuged for one minute at 6000× *g* rpm. Collection tubes containing flow-through was discarded, and the column was inserted in another 2 mL collecting tubes. Then, 500 µL washing buffer I was loaded, followed by centrifuge at 8000× *g* rpm for 1 min. and flow-through was discarded and reinserted the column in the same collecting tubes. Next, 500 µL washing buffer II was added to the column and 38 centrifuged at high speed (≥12,000 × *g* rpm) for 03 min. and collecting tubes along with flow-through was discarded and the column was transferred to 1.5 mL microcentrifuge tubes. Then, 200 µL elusion buffer was loaded to the purification column center, followed by incubation at room temperature for two min. and centrifuged for one min. at 8000× *g* rpm. Purified DNA in microcentrifuge tubes was stored at −20 °C for further use. Purified DNA was subjected to polymerase chain reaction (PCR). For electrophoresis, 0.5 g agarose powder was dissolved in 50 mL of 1 X TAE buffer by heating until a transparent solution was obtained. The gel was stained with ethidium bromide (EB). On cooling, the agarose solution was poured into an electrophoresis gel tray. A comb of appropriate size was fixed in a tray to make wells, and the gel was allowed to solidify at room temperature. DNA samples were mixed with loading dye. DNA samples and primers were loaded into appropriate 39 wells on the gel. Fifty volts is carried out until the counter electrode glass achieves the gel. Ultraviolet short wavelength was used to visualize DNA bands in the gel, and the gel documentation system took photographs. Gel purified DNA and PCR clean-up system (Promega) and quantification was done by using NanoDrop. The amplified product was then sent for sequencing to the Department of Crop Sciences, University of Illinois, Urbana, IL, USA.

polymerase chain reaction (PCR) assay was also carried out for detecting *Streptomyces* species. A DNA product was amplified using the 16S rRNA universal primer (Forward: 5′ CAT TCA CGG AGA GTT TGA TCC 3′; Reverse: 5′ AGA AAG GAG GTG ATC CAG CCC3′) [63]. The procedure was executed by adding 2 μL of DBA template, 5 μL of reaction buffer (10 mM of Tris-HCl, pH 8.8, 50 mM KCL, 3 μL of MgCl_2_, 0.5 μL Taq polymerase, 5 μL mixed dNTP_s_ (2 mM) and primers. The procedure was performed in an automated thermal cycler with that programs as follows: an initial denaturing at 94 °C for minutes, for amplification and annealing 94 °C for 30 s, 46 °C for 40 s, 72 °C for 1 min and extension step 72 °C for 10 min. The PCR product was kept at −40 until loaded onto the gel. After gel electrophoresis, it was pictured under UV light of the gel documentation system [55].

### 4.9. Sequence Analysis

Sequence analysis was also done for molecular studies of *Streptomyces* species, gene annotation and constructing the phylogenetic tree. Moreover, the sequences of the promising *Streptomyces* isolates along with other registered gene sequences retrieved from the GenBank https://www.ncbi.nlm.nih.gov/website and the sequences were aligned using Mega Evolutionary Genetics Analysis (MEGA 7.0.26) software and comparison and sequence similarity were also checked by basic local alignment search tool (BLAST_N_) nucleotide https://blast.ncbi.nlm.nih.gov/Blast.cgi website. Phylogenetic trees were constructed by the neighbor-joining method in MEGA 7.0.26 version with 1000 bootstrap replicates.

### 4.10. Statistical Analysis

The data in the investigation of *S. scabies* pathogenicity and determination of host-disease resistance in potato germplasm were processed by analyzing variance procedures by using IBM SPSS Statistics software. Means were calculated by using Fisher’s protected least significant difference (LSD) procedure at *p* ≤ 0.05. Significant treatment and their combination to control common scab disease were developed using SAS/STAT package. Data were obtained from three independent experiments. In each experiment, three replicates were used.

## 5. Conclusions

The conclusion of the current study is, several species of Streptomyces, which includes S. scabiei, S. acidiscabies, S. europaeiscabiei, S. luridiscabiei, S. niveiscabiei, S. puniciscabiei, S. reticuliscabiei, S. stelliscabiei, S. turgidiscabies, S. ipomoeae were associated with various potato germplasms and efficiently isolated from infected potato tubers. The frequency of diversity and distribution of *Streptomyces* species was closely related to the potato’s different cultivars and checked by screening methods, and resistance was also observed. Environmental factors are conducive to developing common scab disease, which is also favorable for various other species rather than *S. scabies*. Hence, far the genetic diversity of *Streptomyces* species and potato cultivars against these pathogens are still not clear and challenging to scientists. This work also revealed that *S. scabies* is the dominant pathogen on the potato cultivars and cause yield losses. There is a dire need to understand the interaction of *S. scabies* with potato cultivars and other crops in the future. The research findings of the current work will help the breeding of potato cultivars against common scab disease and beneficial for managing *Streptomyces*-associated with potato tubers and disease-free tubers in Punjab, Pakistan. As well as globally where potato is grown as major crop in many countries of the world.

## Figures and Tables

**Figure 1 pathogens-09-00760-f001:**
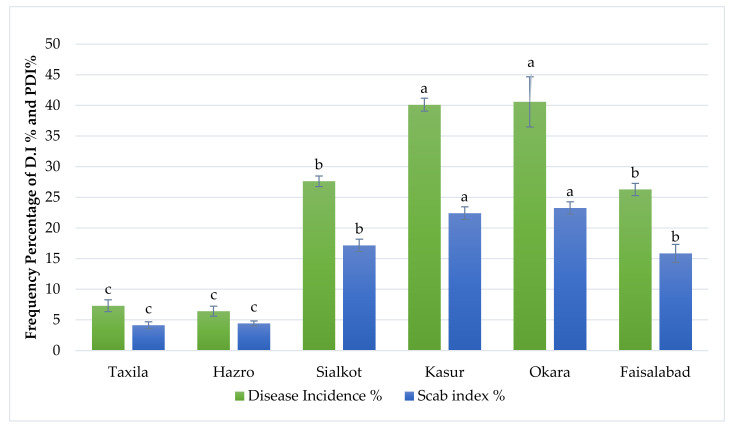
Percent disease incidence and percent scab index in different areas of Pakistan; Taxila City, Rawalpindi; Hazro, Attock; Sialkot; Kasur; Okara; Faisalabad. The means obtained for various treatments with significant differences denoted by different letters, based on Duncan’s multiple test procedure at *p* < 0.05. Disease incidence in Taxila (**c**), Hazro (**c**), Sialkot (**b**), Kasur (**a**), Okara (**a**), Faisalabad (**b**) was observed statistically. Scab index % in Taxila (**c**), Hazro (**c**), Sialkot (**b**), Kasur (**a**), Okara (**a**), Faisalabad (**b**) was also observed statistically.

**Figure 2 pathogens-09-00760-f002:**
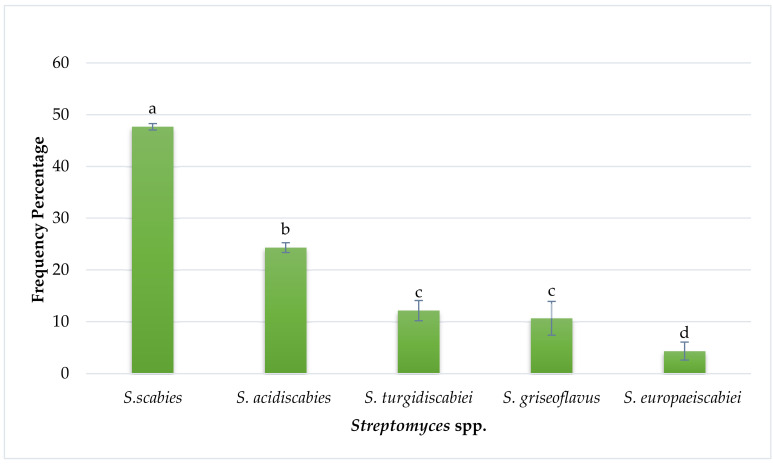
Isolation frequency of *Streptomyces* from potato tubers. *Streptomyces* were isolated and purified on King’s B medium, and the percentage of isolates obtained for each species of *Streptomyces* was calculated by comparison of means. Isolates; *S. scabies* (**a**), *S. acidiscabies* (**b**), *S. turgidiscabiei* (**c**), *S. griseoflavus* (**c**), *S. europaeiscabiei* (**d**) were observed statistically. The means obtained for various treatments with significant differences denoted by different letters, based on Duncan’s multiple test procedure at *p* < 0.05.

**Figure 3 pathogens-09-00760-f003:**
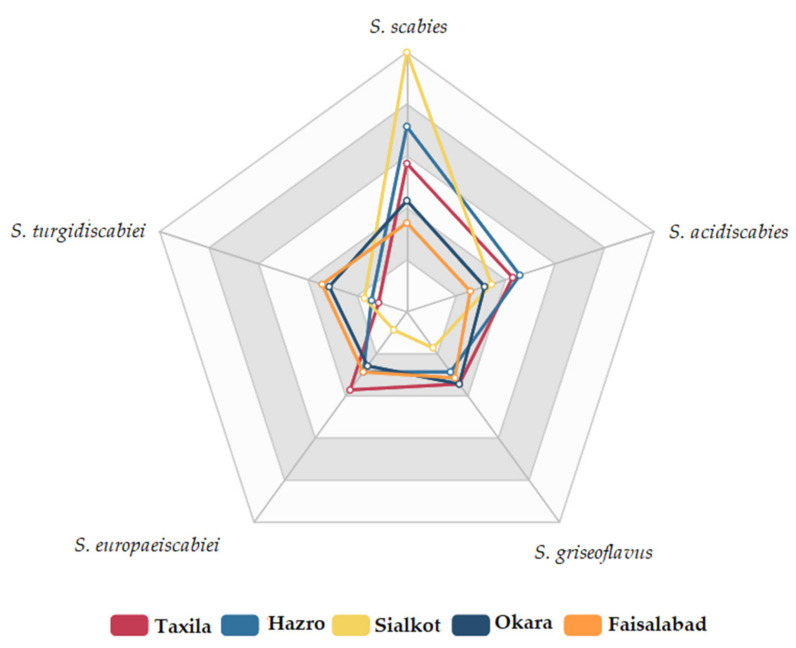
Diversity and distribution of various *Streptomyces* species from samples collected from different potato-growing areas during the 2019 survey.

**Figure 4 pathogens-09-00760-f004:**
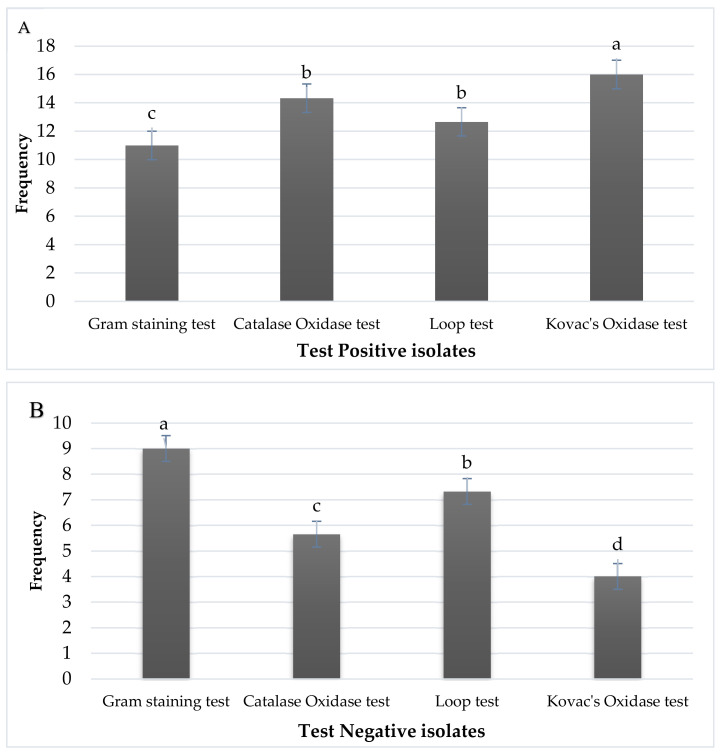
Biochemical characterization of different *Streptomyces* isolates for confirmation. (**A**) Test-positive isolates confirmed by Gram-staining test (c), catalase oxidase test (b), loop test (b) and Kovac’s oxidase test (a) (**B**) Test-negative isolates confirmed by Gram-staining test (a), catalase oxidase test (c), loop test (b) and Kovac’s oxidase test (d). Means obtained for various treatments with significant differences denoted by different letters (a, b, c, d), based on Duncan’s multiple test procedure at *p* < 0.05.

**Figure 5 pathogens-09-00760-f005:**
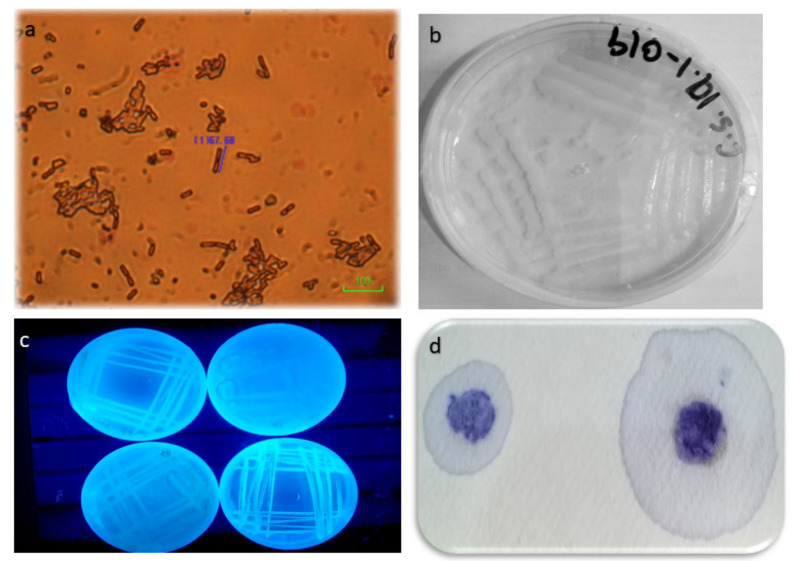
Various biochemical responses of *Streptomyces* spp. (**a**) Microscopy diagram of *S. scabies*; (**b**) Levan test for confirmation of *Streptomyces* spp; (**c**) fluorescent pigment test for the presence of Gram-positive actinomycetes *Streptomyces* (**d**) oxidase test of *Streptomyces.*

**Figure 6 pathogens-09-00760-f006:**
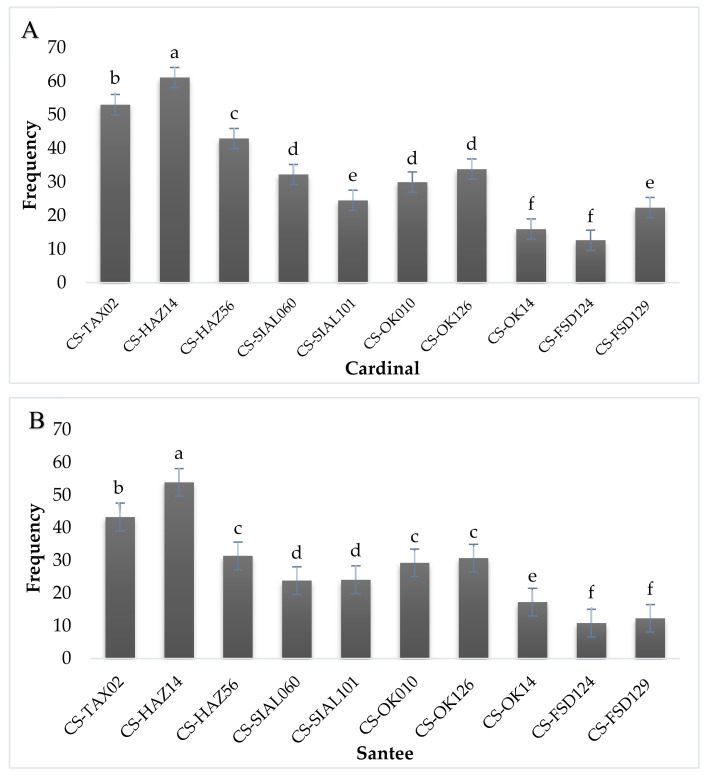
Virulent confirmation/Pathogenicity tests for the confirmation of virulent strains of *Streptomyces* spp. on the potato cultivars. (**A**) Percentage of the mortality rate of *Streptomyces* spp. on the potato cultivar Cardinal; (**B**) percentage mortality rate of *Streptomyces* spp. on the potato cultivar Sante. Means obtained for various treatments with significant differences denoted by different letters, based on Duncan’s multiple test procedure at *p* < 0.05.

**Figure 7 pathogens-09-00760-f007:**
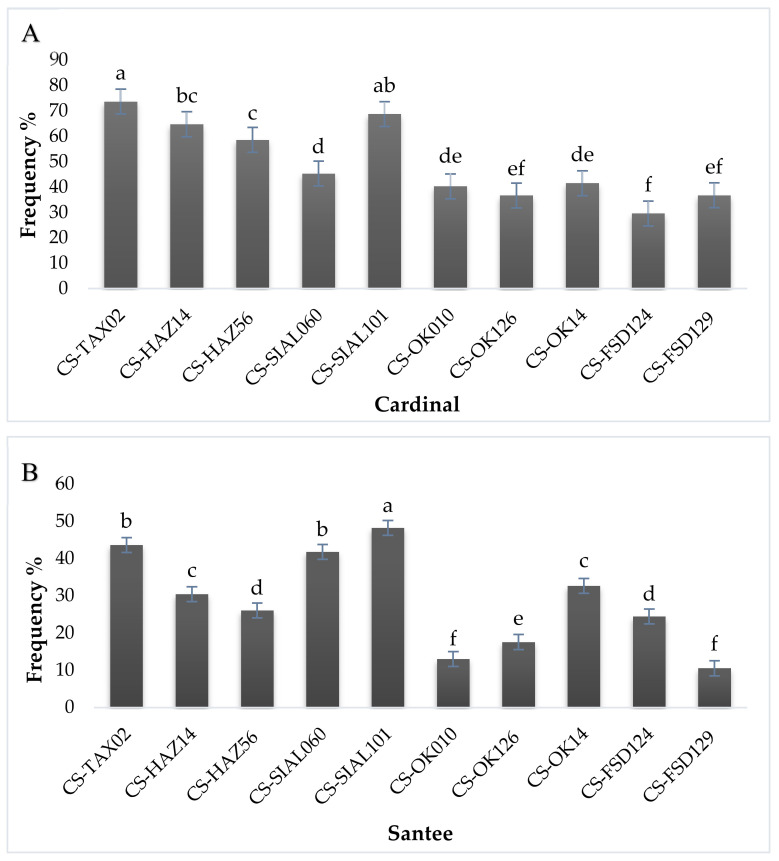
Determination of the Percent tuber area infected in two potato cultivars by pathogenicity assays (pin-prick method). (**A**) Percentage of tuber area infected by *Streptomyces* spp. on the potato cultivar Cardinal, Isolates such as CS-TAX02 (a), CS-HAZ14 (bc), CS-HAZ56 (c), CS-SIAL060 (d), CS-SIAL101 (ab), CS-OK010 (de), CS-OK126 (ef), CS-OK14 (de), CS-FSD124 (f), CS-FSD129 (ef) were observed statistically; (**B**) Percentage tuber area infected by *Streptomyces* spp. on the potato cultivar Santee, Isolates such as CS-TAX02 (b), CS-HAZ14 (c), CS-HAZ56 (d), CS-SIAL060 (b), CS-SIAL101 (a), CS-OK010 (f), CS-OK126 (e), CS-OK14 (c), CS-FSD124 (d), CS-FSD129 (f) were observed statistically. Means obtained for various treatments with significant differences denoted by different letters, based on Duncan’s multiple test procedure at *p* < 0.05.

**Figure 8 pathogens-09-00760-f008:**
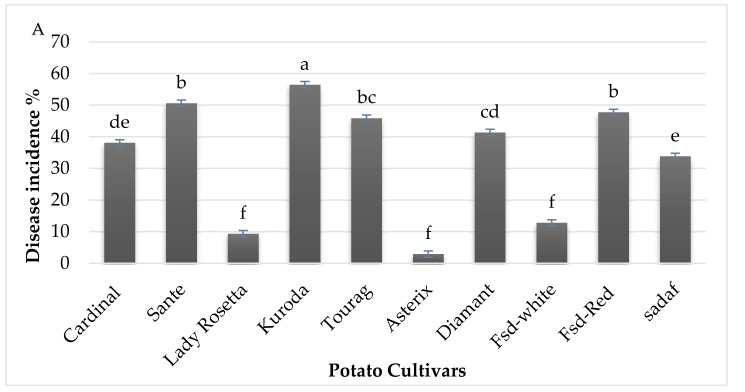
Percent disease incidence and percent scab index of screened potato cultivars against *Streptomyces scabies*. (**A**) Disease incidence of 10 potato cultivars such as Cardinal (de), Santee (b), Lady Rosetta (f), Kuroda (a), Tourag (bc), Asterix (f), Diamant (cd), Fsd-white (f), Fsd-red (b), Sadaf (e) were observed statistically (**B**) Percent scab index of 10 potato cultivars such as Cardinal (b), Santee (a), Lady Rosetta (d), Kuroda (a), Tourag (ab), Asterix (d), Diamant (c), Fsd-white (d), Fsd-red (ab), Sadaf (c) were observed statistically. These parameters were measured based on the disease rating scale of potato scab. The means obtained for various treatments with significant differences are denoted by different letters, based on Duncan’s multiple test procedure at *p* < 0.05.

**Figure 9 pathogens-09-00760-f009:**
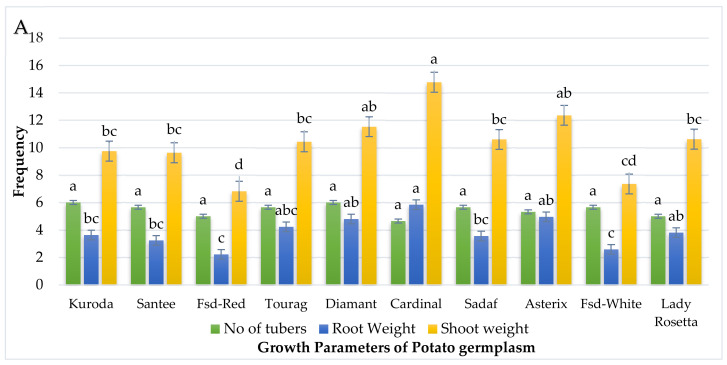
Statistical analysis of the effect of *Streptomyces scabies* on different growth parameters of screened potato cultivars. (**A**) Different growth parameters such as No. of tubers in potato cultivars; Kuroda (a), Santee (a), Fsd-Red (a), Tourag (a), Diamant (a), Cardinal (a), Sadaf (a), Asterix (a), Fsd-white (a), Lady Rosetta (a), Root weight Kuroda (bc), Santee (bc), Fsd-Red (c), Tourag (abc), Diamant (ab), Cardinal (a), Sadaf (bc), Asterix (ab), Fsd-white (c), Lady Rosetta (ab) were observed statistically. Shoot weight in Kuroda (bc), Santee (bc), Fsd-Red (d), Tourag (bc), Diamant (ab), Cardinal (a), Sadaf (bc), Asterix (ab), Fsd-white (cd), Lady Rosetta (bc) were observed statistically. (**B**) Different growth parameters such as Root length for various varieties such as Kuroda (ab), Santee (ab), Fsd-Red (ab), Tourag (a), Diamant (ab), Cardinal (b), Sadaf (ab), Asterix (ab), Fsd-white (ab), Lady Rosetta (a) were observed statistically. Shoot length in Kuroda (ab), Santee (c), Fsd-Red (c), Tourag (c), Diamant (ab), Cardinal (c), Sadaf (c), Asterix (ab), Fsd-white (ab), Lady Rosetta (d) are described statistically. Means obtained for various treatments with significant differences denoted by different letters, based on Duncan’s multiple test procedure at *p* < 0.05.

**Figure 10 pathogens-09-00760-f010:**
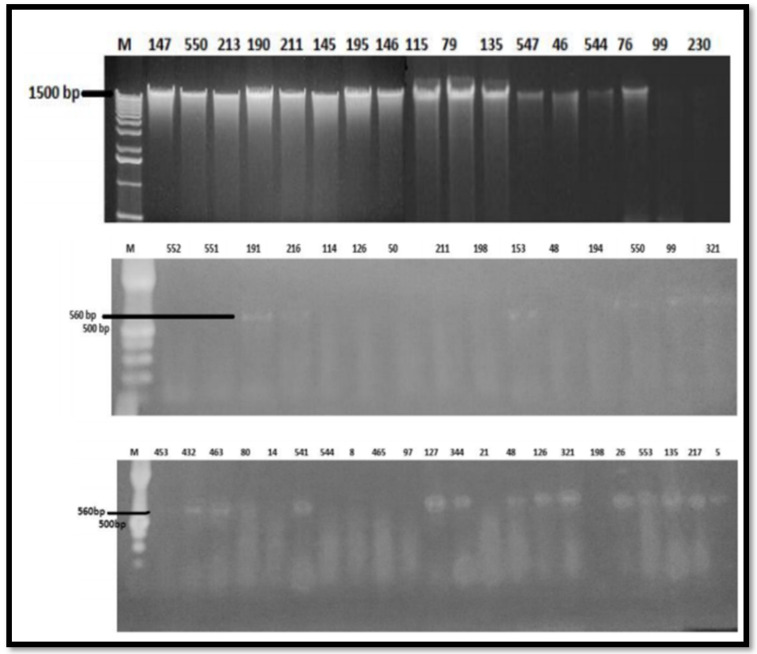
DNA template detected from infected potato tubers and PCR amplification of *Streptomyces scabies*.

**Figure 11 pathogens-09-00760-f011:**
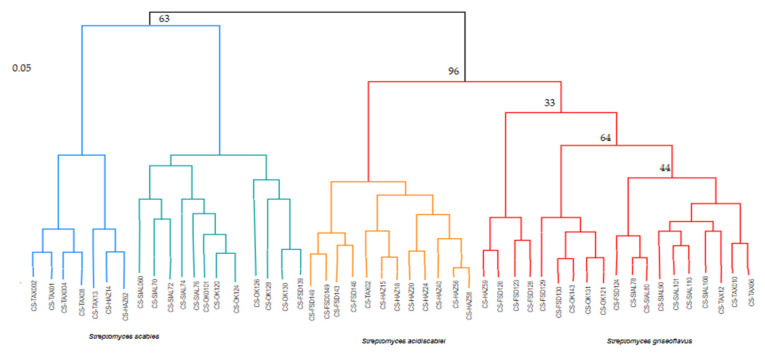
Phylogenetic tree of *Streptomyces* species associated with potato cultivars. The phylogenetic tree was constructed by the maximum-likelihood method by MEGA 7.0.26. Bootstrap support values were obtained from 1000 replications.

**Figure 12 pathogens-09-00760-f012:**
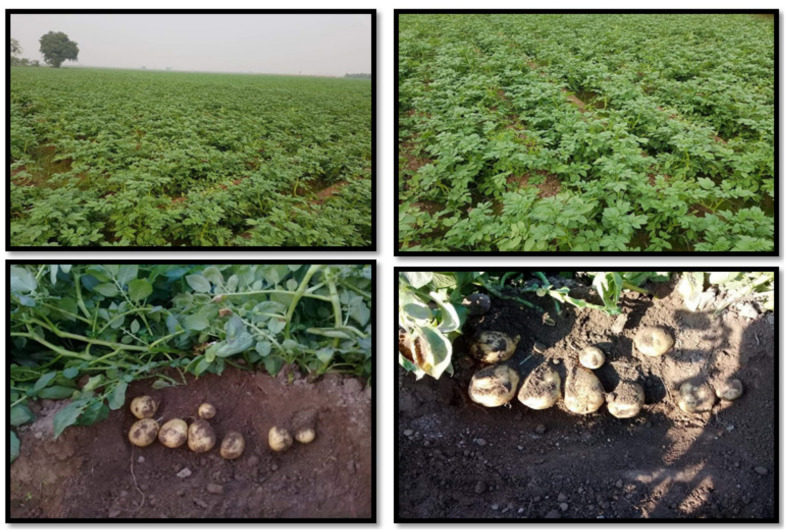
Potato-growing areas of Punjab, Pakistan were visited for sample collection.

**Figure 13 pathogens-09-00760-f013:**
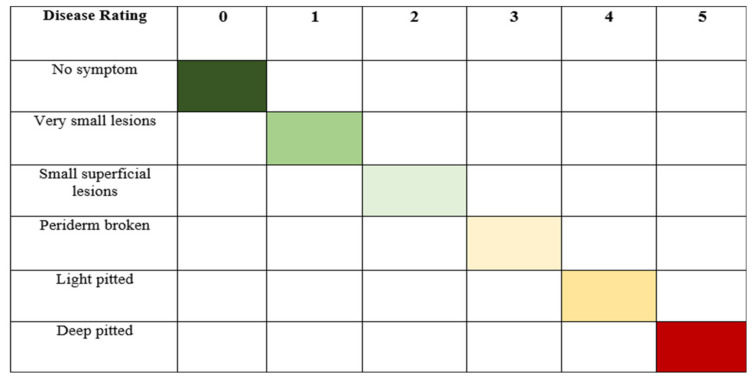
A disease rating scale for the measurement of percent disease incidence of common scab of potato.

**Figure 14 pathogens-09-00760-f014:**
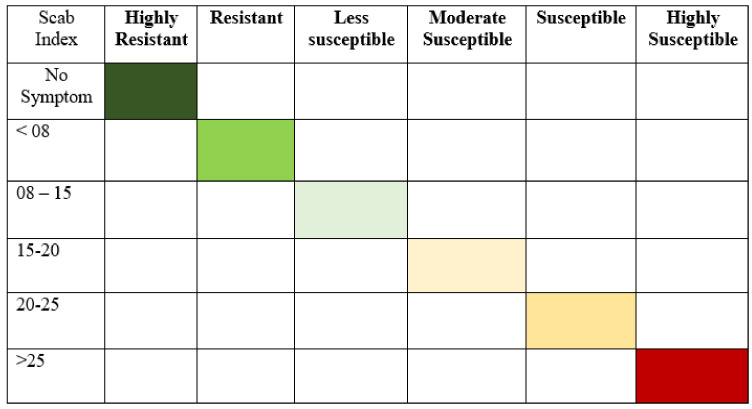
The disease rating scale of percent scab index of common potato scab.

**Table 1 pathogens-09-00760-t001:** Disease percent incidence and percent scab index during the survey of potato fields.

Area	Location	Disease Incidence (%)	Percent Scab Index
Taxila (Rawalpindi)	Thatha	2%	3.1%
	Usman Khattir	3%	3.2%
	Ghari sikandar	16%	6.7%
	Khurum Paracha	12%	5.5%
	Labthatoo	5%	3.6%
Average		8%	4.42%
Hazro (Attock)	Waisa	13%	9.6%
	Mallah	3%	2.6%
	Shamsabad	11%	8.2%
	Sirka	2%	1.9%
	Shadi khan	9%	4.6%
	Kalu kalan	5%	2.5%
Average		7%	4.90%
Sialkot	Kishan garh	22%	13.7%
	Hussain pur	23%	13.9%
	Randhir	34%	18.7%
	Habibpur	41%	21.2%
	Chicharwali	21%	13.2%
Average		28%	16.14%
Kasur	Qutba	40%	22.6%
	Faqeerwala	38%	21.8%
	Dolaywala	42%	23.1%
	Average	40%	22.5%
Okara	Moza Ameer	36%	23.8%
	Salwal	45%	17.5%
	Qadirabad	52%	25.2%
	Average	44.33%	22.16%
Faisalabad	Jaranwala	32%	15.1%
	Ram Diwali	26%	14.9%
	Karamsar	24%	18.7%
	Average	27.33%	16.23%

**Note:** potato scab percent incidence and scab index measured based on the disease rating scales described in materials and methods. These parameters were measured, randomly selected areas in a zigzag manner.

**Table 2 pathogens-09-00760-t002:** Brief information on *Streptomyces* spp. isolated from different potato germplasm and growing areas of Punjab, Pakistan.

Isolates	Potato Cultivars	Location of Sampling	Suggested Identification
CS-TAX01	Cardinal	Taxila (Rawalpindi)	*Streptomyces scabies*
CS-TAX02	Cardinal	Taxila (Rawalpindi)	*Streptomyces acidiscabies*
CS-TAX002	Asterix	Taxila (Rawalpindi)	*Streptomyces scabies*
CS-TAX004	Lady Rosetta	Taxila (Rawalpindi)	*Streptomyces scabies*
CS-TAX06	Lady Rosetta	Taxila (Rawalpindi)	*Streptomyces griseoflavus*
CS-TAX08	Faisalabad red	Taxila (Rawalpindi)	*Streptomyces scabies*
CS-TAX010	Faisalabad red	Taxila (Rawalpindi)	*Streptomyces griseoflavus*
CS-TAX12	Faisalabad red	Taxila (Rawalpindi)	*Streptomyces griseoflavus*
CS-TAX13	Faisalabad white	Taxila (Rawalpindi)	*Streptomyces scabies*
CS-HAZ14	Faisalabad white	Hazro (Attock)	*Streptomyces scabies*
CS-HAZ15	Sadaf	Hazro (Attock)	*Streptomyces acidiscabies*
CS-HAZ18	Sadaf	Hazro (Attock)	*Streptomyces acidiscabies*
CS-HAZ20	Sadaf	Hazro (Attock)	*Streptomyces acidiscabies*
CS-HAZ24	Sadaf	Hazro (Attock)	*Streptomyces acidiscabies*
CS-HAZ40	Sadaf	Hazro (Attock)	*Streptomyces acidiscabies*
CS-HAZ56	Sadaf	Hazro (Attock)	*Streptomyces acidiscabies*
CS-HAZ58	Tourag	Hazro (Attock)	*Streptomyces acidiscabies*
CS-HAZ59	Tourag	Hazro (Attock)	*Streptomyces acidiscabies*
CS-HAZ62	Tourag	Hazro (Attock)	*Streptomyces scabies*
CS-SIAL060	Tourag	Sialkot	*Streptomyces scabies*
CS-SIAL70	Diamant	Sialkot	*Streptomyces scabies*
CS-SIAL72	Diamant	Sialkot	*Streptomyces scabies*
CS-SIAL74	Diamant	Sialkot	*Streptomyces scabies*
CS-SIAL76	Diamant	Sialkot	*Streptomyces scabies*
CS-SIAL78	Diamant	Sialkot	*Streptomyces griseoflavus*
CS-SIAL80	Diamant	Sialkot	*Streptomyces griseoflavus*
CS-SIAL90	Diamant	Sialkot	*Streptomyces griseoflavus*
CS-SIAL101	Kuroda	Sialkot	*Streptomyces griseoflavus*
CS-SIAL110	Kuroda	Sialkot	*Streptomyces griseoflavus*
CS-OK108	Kuroda	Okara	*Streptomyces griseoflavus*
CS-OK0101	Kuroda	Okara	*Streptomyces scabies*
CS-OK120	Kuroda	Okara	*Streptomyces scabies*
CS-OK124	Asterix	Okara	*Streptomyces scabies*
CS-OK126	Asterix	Okara	*Streptomyces scabies*
CS-OK128	Asterix	Okara	*Streptomyces scabies*
CS-OK130	Faisalabad red	Okara	*Streptomyces scabies*
CS-OK143	Faisalabad white	Okara	*Streptomyces griseoflavus*
CS-OK131	Faisalabad white	Okara	*Streptomyces griseoflavus*
CS-OK121	Diamant	Okara	*Streptomyces griseoflavus*
CS-FSD124	Sadaf	Faisalabad	*Streptomyces griseoflavus*
CS-FSD126	Sadaf	Faisalabad	*Streptomyces acidiscabies*
CS-FSD123	Diamant	Faisalabad	*Streptomyces acidiscabies*
CS-FSD128	Kuroda	Faisalabad	*Streptomyces acidiscabies*
CS-FSD129	Lady Rosetta	Faisalabad	*Streptomyces acidiscabies*
CS-FSD130	Lady Rosetta	Faisalabad	*Streptomyces acidiscabies*
CS-FSD139	Sante	Faisalabad	*Streptomyces scabies*
CS-FSD149	Sante	Faisalabad	*Streptomyces scabies*
CS-FSD0149	Sante	Faisalabad	*Streptomyces scabies*
CS-FSD143	Diamant	Faisalabad	*Streptomyces scabies*
CS-FSD146	Diamant	Faisalabad	*Streptomyces scabies*

**Table 3 pathogens-09-00760-t003:** Cultural characteristics of *Streptomyces* species.

Culture Media	Growth	Colony Color	Aerial Mycelium
nutrient agar	Moderate	Creamy White	Abundant
Potato dextrose agar	Moderate	Creamy White	Abundant
Yeast malt agar	Well Abundant	Creamy White	Well Abundant
King’s B media	Moderate	White to Creamy	Poor
Oat meal agar	Moderate	White	Abundant
Potato yeast extract	Moderate	Brown	Abundant
Czapek media	Moderate	White	Abundant

**Table 4 pathogens-09-00760-t004:** Morphologic characteristics of *Streptomyces* species.

Isolates	Spore Shape	Spore Color	Septations	Spore Size (μm)	
				Length (Min–Max) Mean	Width(Min–Max) Mean
CS-TAX01	Pointed from one end and blunted from another end	Hyaline	2 celled actinomycetes conidia	(9–18) 13.5	(5–6) 5.5
CS-TAX002	Pointed from one end and blunted from other end	Hyaline	2 celled actinomycetes conidia	(9–15) 12	(5–6) 5.5
CS-TAX004	Pointed from one end	Hyaline	2 celled actinomycetes conidia	(9–20) 14.5	(6–7) 6.5
CS-TAX08	Pointed from one end	Hyaline	2 celled actinomycetes conidia	(9–17) 13	(5–7) 6
CS-TAX13	Pointed from one end and blunted from other end	Hyaline	2 celled actinomycetes conidia	(9–20) 14.5	(6–7) 6.5
CS-HAZ14	Pointed from one end	Hyaline	2 celled actinomycetes conidia	(10–20) 15	(6–7) 6.5
CS-HAZ62	Pointed from one end	Hyaline	2 celled actinomycetes conidia	(10–21) 15.5	(5–7) 6
CS-SIAL060	Pointed from one end	Hyaline	Single-cell actinomycetes conidia	(8–22) 15	(3–7) 5
CS-SIAL70	Pointed from one end and blunted from other end	Falcate	Single-cell actinomycetes conidia	(13–25) 14	(4–7) 5.5
CS-SIAL72	Pointed from one end and blunted from other end	Fusiform	Single-cell actinomycetes conidia	(12–22) 17	(6–8) 7
CS-SIAL74	Pointed from two ends	Falcate	2 celled actinomycetes conidia	(15–21) 18	(2–9) 5.5
CS-SIAL76	Pointed from two ends	Hyaline	2 celled actinomycetes conidia	(13–17) 15	(5–8) 6.5
CS-OK0101	Pointed from two ends	Hyaline	Single-cell actinomycetes conidia	(13–21) 17	(5–10) 7.5
CS-OK120	Pointed from one end and blunted from other end	Hyaline	Single-cell actinomycetes conidia	(16–17) 16.5	(4–7) 5.5
CS-OK124	Pointed from one end and blunted from other end	Falcate	Single-cell actinomycetes conidia	(15–25) 20	(5–9) 7
CS-OK126	Pointed from two ends	Falcate	Single-cell actinomycetes conidia	(12–19) 15.5	(6–7) 6.5
CS-OK128	Pointed from two ends	Falcate	2 celled actinomycetes conidia	(15–22) 18.5	(3–6) 4.5
CS-OK130	Pointed from two ends	Hyaline	2 celled actinomycetes conidia	(14–23) 13.5	(2–7) 4.5
CS-FSD139	Pointed from one end and blunted from other end	Hyaline	2 celled actinomycetes conidia	(16–25) 20.5	(3–10) 6.5
CS-FSD149	Pointed from two ends	Fusiform	2 celled actinomycetes conidia	(19–20) 19.5	(6–7) 6.5
CS-FSD0149	Pointed from one end	Fusiform	2 celled actinomycetes conidia	(20–22) 21	(2–6) 4
CS-FSD143	Pointed from one end	Fusiform	Single-cell actinomycetes conidia	(15–18) 16.5	(7–7) 7
CS-FSD146	Pointed from one end	Hyaline	Single-cell actinomycetes conidia	(16–17) 16.5	(5–7) 6

**Table 5 pathogens-09-00760-t005:** Percent scab incidence, percent scab index and disease reaction of common scab disease in the field area of the department of plant pathology, PMAS—arid agriculture university, Rawalpindi (Pakistan).

S. No.	Variety	Percent Scab Incidence	Percent Scab Index	Reaction
1	Cardinal	37.46	15.91	Medium susceptible
2	Santee	49.63	20.5	Susceptible
3	Lady Rosetta	17.55	5.90	Less susceptible
4	Kuroda	56.35	20.65	Susceptible
5	Tourag	46.25	18.77	Medium susceptible
6	Asterix	2.90	1.47	Resistant
7	Diamant	41.55	11.38	Less susceptible
8	Faisalabad White	22.53	1.08	Resistant
9	Faisalabad red	75.24	25.18	Highly susceptible
10	Sadaf	33.38	7.09	Less susceptible
11	Control	12.01	6.00	Resistant

**Table 6 pathogens-09-00760-t006:** LSD all-pairwise comparisons test to check the effect of *Streptomyces scabies* on percent disease incidence, percent scab index and No. of tubers of potato.

Varieties	Percent Disease Incidence	Percent Scab Index	No. of Tubers
Kuroda	56.48 a	20.71 ab	6.01 a
Santee	50.61 b	21.16 a	5.66 a
Faisalabad red	47.75 c	20.64 ab	5.01 a
Tourag	45.91 c	18.56 bc	5.66 a
Diamant	41.38 d	11.13 d	6.01 a
Cardinal	38.11 e	16.51 c	4.66 a
Sadaf	33.83 f	7.09 e	5.66 a
Asterix	2.94 g	1.55 f	5.33 a
Faisalabad white	2.81 g	1.25 f	5.66 a
Lady Rosetta	2.35 g	1.05 f	5.01 a

**Note:** Data are means from three independent replicates. Different lowercase letters in the same column show a significant difference at *p* < 0.05 by LSD and Duncan’s multiple range test.

**Table 7 pathogens-09-00760-t007:** LSD all-pairwise comparisons test to check the effect of *Streptomyces scabies* on various growth parameters of potato.

Varieties	Root Weight (g)	Shoot Weight (g)	Root Length (cm)	Shoot Length (cm)
Kuroda	3.64 bc	9.76 bc	20.66 ab	30.66 c
Santee	3.24 bc	9.64 bc	19.66 ab	39.66 ab
Faisalabad red	2.22 c	6.83 d	19.66 ab	41.01 ab
Tourag	4.24 abc	10.44 bc	21.33 a	41.33 ab
Diamant	4.81 ab	11.54 ab	19.01 ab	34.01 bc
Cardinal	5.85 a	14.78 a	16.01 b	40.33 ab
Sadaf	3.56 bc	10.61 bc	16.33 ab	40.01 ab
Asterix	4.96 ab	12.37 ab	20.33 ab	26.66 c
Faisalabad white	2.59 c	7.36 cd	18.33 ab	30.66 c
Lady Rosetta	3.81 ab	10.63 bc	21.33 a	46.01 a

**Note:** Data are means from three independent replicates. Different lowercase letters in the same column show a significant difference at *p* < 0.05 by LSD and Duncan’s multiple range test.

**Table 8 pathogens-09-00760-t008:** Identification and similarity of *Streptomyces* isolates with reference accession numbers from NCBI.

Isolates	Identified Species	Similarity%	Accession Number
CS-TAX01	*Streptomyces scabies*	100%	M57297.1
CS-TAX002	*Streptomyces scabies*	98%	AF031232.1
CS-TAX004	*Streptomyces scabies*	100%	AM293590.1
CS-TAX08	*Streptomyces scabies*	99%	HM018077.1
CS-TAX13	*Streptomyces griseoflavus*	96%	JX284407.1
CS-HAZ14	*Streptomyces europaeiscabiei*	90%	HQ441817.1
CS-HAZ62	*Streptomyces europaeiscabiei*	89%	AY207595.1
CS-SIAL060,	*Streptomyces scabies*	99%	AY207602.1
CS-SIAL70	*Streptomyces scabies*	98%	AB301479.1
CS-SIAL72	*Streptomyces scabies*	99%	Y15497.1

**Table 9 pathogens-09-00760-t009:** Collection of potato germplasm from different sources.

S. No	Variety Name	Location/Source
1	Cardinal	National agricultural research institute (NARC), Islamabad
2	Santee	National agricultural research institute (NARC), Islamabad
3	Lady Rosetta	National agricultural research institute (NARC), Islamabad
4	Kuroda	National agricultural research institute (NARC), Islamabad
5	Tourag	National agricultural research institute (NARC), Islamabad
6	Asterix	Potato Research Institute (PRI), Sahiwal
7	Diamant	Potato Research Institute (PRI), Sahiwal
8	Faisalabad white	Potato Research Institute (PRI), Sahiwal
9	Faisalabad red	Potato Research Institute (PRI), Sahiwal
10	Sadaf	Potato Research Institute (PRI), Sahiwal

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
