# Peer review of "Investigation of Streptomyces scabies Causing Potato Scab by Various Detection Techniques, Its Pathogenicity and Determination of Host-Disease Resistance in Potato Germplasm"

_pathogens, 2020, doi:10.3390/pathogens9090760_

Round 1
Reviewer 1 Report
In this paper Zheng et. al describe several different Streptomyces species that cause potato scab and determine different potato cultivars that are scab resistant. Overall I found the science and methods used to be sound and robust. However, the English and grammar leave a lot to be desired. The spelling and grammar mistakes are too numerous to include in the review.
There were a couple of points in the introduction where antibiotics were very randomly brought up. These should either be removed or worked into the introduction in a more logical way.
In the paragraph starting at line 97 there is some redundancy. The use of disease resistant cultivars is mentions multiple times in the same paragraph.
In the text of the manuscript table 4 is discussed before table 3. The order of the tables should be switched.
Otherwise, English excepted, the manuscript is well written and easy to follow. All table and figures are labeled and explained well.
Reviewer 2 Report
Dear Authors,
I have great opportunity to revive research manuscript entitled: ”Investigation of Streptomyces scabies causing potato scab by various detection techniques, its pathogenicity and determination of host-disease resistance in potato germplasm” (manuscript id: pathogens-920674) which is considered for publication in Pathogens journal. I analyze whole manuscript and is generally well written and outline interesting new findings about different species of Streptomyces. However, way of presentation of results (especially in graphic aspect) must be much improved because now many results is unclear and misleading. Also some elements of methodology must be clarified. Therefore because some amount of work needed to improve manuscript I suggest major revision. Reasons for that decision I present in a form of list of specific comments presented below:
General comment:
Article is focused on major of situation Punjab, Pakistan. However I would like strongly suggest to connect authors results with some probable aspects/future aspects information and knowledge for the resistance against common scab disease and also disease free cultivars and disease management worldwide not only in one region.
Introduction section:
Generally well written section. However statement (line 106-107) : “Although the significance of Streptomyces species to potato crop has been documented in previous studies…” is not clear. Authors have in mind their results from previous publication or publication of other research teams. In both cases in this point previous results should have appropriate citation.
Results section:
Why figures and table numbers in main text of articles is made with color font? This make all figure citation and also literature citation prepared not exactly journal publication rules. Figure 1 is not clear at all. This chart must be prepared with use of bars with statistical assessment of difference between Disease Incidence (%) and Scab index %. Currently Figure 1 is difficult to read in context results comparison. Figure 2 is also problematic because it not present scientifically performed statistical analyses. What’s worse in description of this figure authors add information “The means obtained for various treatments with significant differences, which are denoted by different letters, based on Duncan’s multiple test procedure at p<0.05”. No letters are marked on this figure. Authors must make Figure 2 bigger (better quality) with appropriate markings of statistical significant values. This need to be done because some of values on Figure 2 is not statistically different from others. On Figure 4A some values is not significantly different but they area mark like one. Catalase Oxidase test and loop test is not statistically different so the must be marked b not b and c. In Figure 5 authors must add better (hd qulaity) photo Gram staining test. Now it Is not good for publication. I suggest photo from microscope or binocular. Figure 9 is too small and too low quality to read results . And again authors claims that results was assessed statistically (in description of figure 9) but any significant differences, which are not denoted by different letters. So this figure whole must be much improved. I am sorry but Figure 10 has so many pixels that is difficult to observed presented DNA template.
Discussion section:
This part is too short in context of amount of presented results. It makes manuscript not well balanced in context of all sections.
Materials and Methods section:
Authors should add information about number of plants/tubers of each variety was collected form the fields. It is crucial for all calculated mean values. Authors add many information but no data about number of collected plant/tubers from each variety from each field. Authors must also outline the age of tubers. Because it is crucial in context disease severity measured on the basis of symptoms on the surface of tubers. If tubers are younger the could have only epidermis not full developed Periderm. How authors was able to assessed symptoms on surface without firm statement about age of tubers. If this fact was not checked then susceptibility to pathogen could not be compared. Because in some cases tuber could have epidermis and in other periderm and resistance for pathogen is different with presence of various covering tissues.
Sincerely,
Author Response
Dear Sir. Please see the attachment.

Round 2
Reviewer 2 Report
Dear Authors,
All my suggestions were added.
Sincerely,